# Impact of synonymous mutations in the $bla_{TEM-3}$ gene on gene expression and *Escherichia coli* fitness

Rinku Dhungana,[1] Heba Kaadan,[1] Aparna Paudel,[1] Peter Oelschlaeger[1]

**ABSTRACT** Synonymous mutations (SMs) have a significant effect on the expression of the β-lactamase TEM-3 and the resistance phenotype conferred. Cell-free transcription did not reproduce this effect. The amount of mRNA obtained was identical for $bla_{TEM-3}$ transcripts with or without SMs, and the transcripts were indistinguishable in terms of secondary structure and thermal de- and renaturation, as shown by circular dichroism experiments. In asymmetric competition assays, cells expressing the gene with SMs replaced cells expressing the gene without SMs after five days of passaging in a low concentration of ceftazidime. However, without antibiotic present, cells expressing the gene without SMs replaced cells expressing the gene with SMs. These experiments indicate that either post-transcriptional or *in cellulo* processes are responsible for the differences in expression levels and fitness. Possible mechanisms to be tested in future studies include epigenetic effects (different degrees of methylation) and the removal of the gene in the absence of β-lactam by recombination. Although the phagemid used in this study has the ampicillin resistance gene removed and replaced with a chloramphenicol resistance gene, the 5′-terminal portion of the $bla_{TEM-1a}$ gene is still present and might account for deletion by recombination. The question would then become why recombination is observed more in the phagemids with the gene without SMs.

**IMPORTANCE** Most studies focusing on the evolution of antibiotic resistance focus on nonsynonymous mutations (NMs) in genes encoding proteins that cause resistance, such as β-lactamases. NMs result in amino acid changes that are often responsible for an extended substrate spectrum, insensitivity to inhibitors, or suppression of destabilizing effects introduced by other NMs. Although these mutations are very important, comparisons of natural sequences as well as directed evolution experiments have also identified synonymous mutations (SMs), but their impact is rarely studied. Here, we provide evidence that SMs in a β-lactamase gene alter the fitness levels conferred to *Escherichia coli* cells. Our results suggest that transcription efficiency is not affected, but that epigenetic factors could affect the stability of the gene inside the bacteria. Translation and protein folding efficiency could also not be ruled out thus far. Studying SMs will improve our understanding of how antibiotic resistance evolves and help us combat it.

**KEYWORDS** gene deletion, gene expression, beta-lactamase, antibiotic resistance, synonymous mutation

In a previous report about synonymous mutations (SMs) in three genes encoding TEM β-lactamase variants, SMs had the greatest effect in $bla_{TEM-3}$ (1). TEM-3 expression confers the 2be phenotype, which is characterized by resistance to extended-spectrum β-lactams, such as third-generation cephalosporins and aztreonam (2, 3). Western blots indicated TEM-3 expression in *Escherichia coli* DH10B cells from its published gene (4), which includes four SMs relative to the $bla_{TEM-1a}$ gene (5) (Fig. S1 in the Supplemental

Address correspondence to Peter Oelschlaeger, poelschlaeger@westernu.edu.

The authors declare no conflict of interest.

Material), was more than fourfold higher than when it was expressed from an artificial gene containing only nonsynonymous mutations (NMs). The former is designated as the $bla_{TEM-3}$(+sm) gene and the latter as the $bla_{TEM-3}$(-sm) gene. In disc diffusion assays with *E. coli* cells expressing these genes, $bla_{TEM-3}$(+sm) resulted in smaller diameters than the $bla_{TEM-3}$(-sm) gene. The minimum inhibitory concentration (MIC) of ceftazidime was 64 times higher with the $bla_{TEM-3}$(+sm) gene, while no significant difference was observed with the other antibiotics tested.

Two avenues were pursued to decipher the underlying mechanisms responsible for the different expression and resistance levels in *E. coli*. Firstly, cell-free transcription and analysis of the transcript by circular dichroism (CD) spectropolarimetry were carried out to discern if transcription efficiency or stability of the transcript was affected. Cell-free transcription (MEGAscript Transcription Kit, ThermoFisher) did not result in different amounts of transcript (Fig. 1A), and the transcripts appeared to degrade at equal rates as assessed by agarose gel electrophoresis of transcript after six months of storage at −20°C and several freeze-thaw cycles (Fig. 1A). CD scans of the two transcripts at 25 and 100°C, respectively, are superimposable, indicating no difference in secondary structure (Fig. 1B), and thermal denaturation measured by CD at 270 nm was identical and completely reversible (Fig. 1C). Cell-free translation from these transcripts and coupled transcription/translation was attempted but did not yield detectable amounts of TEM-3 possibly due to insufficient formation of a disulfide bond present in TEMs (6) and improper protein folding.

Another avenue is to study the cellular background in more detail. Harboring vectors with the different genes could affect growth independent of antibiotics. Growth curves of *E. coli* DH10B cells harboring either pBC SK(+)-$bla_{TEM-3}$(-sm) or pBC SK(+)-$bla_{TEM-3}$(+sm), an empty pBC SK(+) vector, or no vector, in Müller-Hinton broth (MHB) showed no significant difference in growth (data not shown). To assess the fitness conferred by either expression of $bla_{TEM-3}$(-sm) or $bla_{TEM-3}$(+sm) in the presence of a small concentration (31 ng/mL or $10^{-9}$ times MIC) of ceftazidime, an asymmetric competition (AC) assay (7) was adapted. In short, starting with mixtures of cells expressing either the -sm or +sm genes in ratios of 1:100 or 100:1, cultures were grown in MHB in the presence of 31 ng/mL ceftazidime and passaged once per day for five days (1:50 dilution). At the end of the assay, cells were diluted and plated on agar plates (200 colonies target, actual ~50–500 colonies). Ten colonies from each plate were randomly isolated, their plasmids were extracted, and the $bla_{TEM-3}$ genes were sequenced. Under these conditions, cells expressing $bla_{TEM-3}$(-sm) started at 1% never replaced cells expressing $bla_{TEM-3}$(+sm). In contrast, cells expressing $bla_{TEM-3}$(+sm) starting at 1% constituted ~23% of colonies isolated at the end of the experiment (Fig. 2A). Interestingly, when agar plates with colonies were kept at 4°C for four weeks and then used in the same assay, cells expressing $bla_{TEM-3}$(+sm) constituted almost all (~97%) of the colonies isolated. To test if the greater fitness of cells expressing $bla_{TEM-3}$(+sm) was due to ceftazidime resistance, the same experiment was repeated without antibiotic

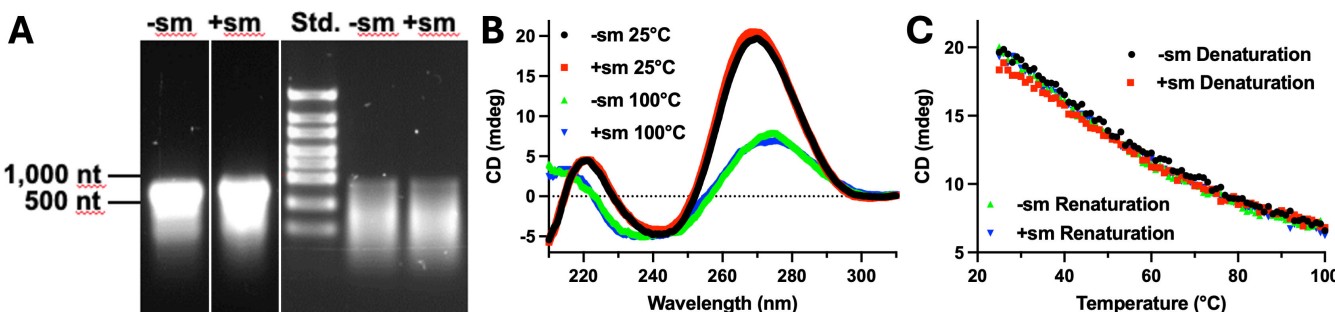

**FIG 1** (A) Agarose gel electrophoresis of transcripts obtained after cell-free transcription of the -sm and +sm genes (left side), RNA standard in the middle, and transcripts after storage at −20°C for six months and several freeze-thaw cycles (right side). (B) Circular dichroism (CD) scans of the -sm and +sm transcripts at 25°C and 100°C. (C) Thermal denaturation (from 25°C to 100°C) and renaturation (from 100°C to 25°C) of the two transcripts measured by CD signal at 270 nm.

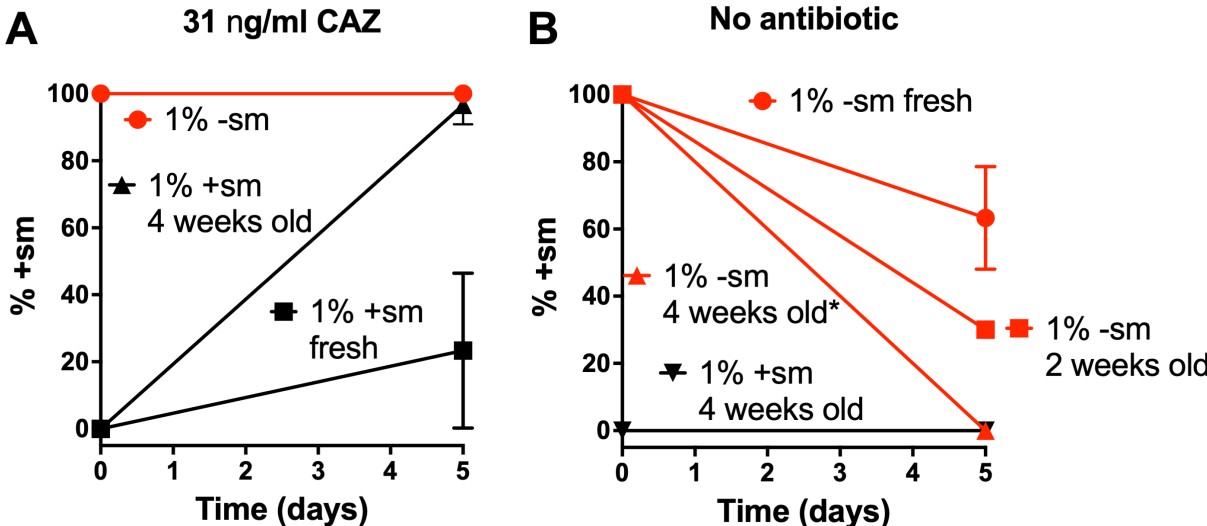

**FIG 2** Results of asymmetric competition assays with ceftazidime (A) and without antibiotic (B). At day 0, the assays were started with either 1% -sm and 99% + sm cells or vice versa. At day 5, cells were harvested and plated on agar plates. Ten colonies were randomly isolated and identified based on their $bla_{TEM-3}$ sequence. All experiments were carried out three times except the experiment with 2-week-old cells in panel B, which was done once. Labels of the lines in the graphs indicate the initial conditions. The results for 1% -sm in panel A and 1% + sm in panel B (horizontal lines) represent 6 (3 with fresh and 3 with 4-week-old cells) and 7 (3 with fresh, 1 with 2-week-old, and 3 with 4-week-old cells) experiments, respectively. Means ± standard deviations are shown. *In some of these cells, the -sm gene was deleted as explained in the text.

present (Fig. 2B). This resulted in the reverse outcome. Cells expressing the +sm gene at an initial 1% never replaced cells expressing the -sm gene. With freshly transformed cells, about 37% of colonies picked at the end of the experiment that started with 1% -sm cells contained the plasmid encoding the -sm gene. With cells from agar plates kept at 4°C for two and four weeks, but otherwise the same conditions, this percentage increased to 70% and 100%, respectively. In the absence of antibiotics, the increased expression level of TEM-3 from the $bla_{TEM-3}$(+sm) seems to lower fitness. Some of the cells also contained genes truncated after 210 nucleotides with an a109c NM, encoding a fragment with Q39—as in TEM-1—rather than K39, which is found in TEM-3. Upon closer examination, these truncated genes seem to be relics of the partially removed $bla_{TEM-1a}$ gene that occurred when pBC SK(+) was generated from pBluescript II (Agilent Technologies) and the $bla_{TEM-3}$(-sm) gene was deleted, likely through homologous recombination.

To investigate this phenomenon further, Epsilometer tests (E-tests) (8) were performed. The results with old cells (Fig. S2A) resemble previous findings from disc diffusion assays (1), and the results for the -sm cells were comparable to the negative controls harboring the same plasmid without $bla_{TEM-3}$ genes. Cells from the agar plates used in these tests were isolated, their plasmids extracted, and the $bla_{TEM-3}$ genes sequenced. All the -sm plasmids had the -sm gene removed, as described above for the AC assay, while the +sm plasmids were intact. The experiment was repeated with freshly transformed cells selected for chloramphenicol resistance encoded on pBC SK(+) plus carbenicillin to ensure full-length $bla_{TEM-3}$ genes. In these E-tests, the -sm cells were comparable to the +sm cells (Fig. S2B). Sequencing of the plasmids harbored by these cells confirmed that the full-length $bla_{TEM-3}$ genes were still present. Further studies on why $bla_{TEM-3}$(-sm) is apparently more prone to deletion than $bla_{TEM-3}$(+sm) or $bla_{TEM-1a}$ are underway.

These results confirm that SMs in the $bla_{TEM-3}$ gene increase antibiotic resistance (1). Cell-free transcription demonstrates that neither transcription efficiency nor transcript stability is responsible for this effect. The facts that the transcripts are indistinguishable in terms of secondary structure and thermal stability, as assessed by CD, and that no rare codons are used (1) suggest that translation is also unlikely to be a limiting factor. The -sm gene in the pBC SK(+) plasmids seems to be less stable than the +sm gene

when there is no β-lactam selection. This could be due to epigenetic effects (9, 10). In this context, it is interesting to note that both $bla_{TEM-1a}$ and $bla_{TEM-3}$(+sm) contain seven 5′-gatc-3′ sites for methylation by DNA adenine methyltransferase (Dam), while $bla_{TEM-3}$(-sm) only contains six. Future studies will investigate the molecular mechanisms underlying the differential deletion of the -sm versus +sm $bla_{TEM-3}$ genes, as well as the effects of SMs in cells and/or vectors that eliminate possible recombination.

## AUTHOR AFFILIATION

[1]Department of Biotechnology and Pharmaceutical Sciences, College of Pharmacy, Western University of Health Sciences, Pomona, California, USA

## PRESENT ADDRESS

Rinku Dhungana, Department of Biological Sciences, Kenneth P. Dietrich School of Arts and Sciences, University of Pittsburgh, Pittsburgh, Pennsylvania, USA

## AUTHOR ORCIDs

Rinku Dhungana ⓘ http://orcid.org/0009-0006-3035-3116
Heba Kaadan ⓘ http://orcid.org/0009-0000-5283-7396
Aparna Paudel ⓘ http://orcid.org/0000-0003-0160-7296
Peter Oelschlaeger ⓘ http://orcid.org/0000-0001-5949-9297

## AUTHOR CONTRIBUTIONS

Rinku Dhungana, Investigation, Methodology | Heba Kaadan, Investigation, Methodology | Aparna Paudel, Investigation, Methodology | Peter Oelschlaeger, Conceptualization, Data curation, Formal analysis, Project administration, Resources, Supervision, Visualization, Writing – original draft, Writing – review and editing

## ADDITIONAL FILES

The following material is available online.

### Supplemental Material

**Supplemental figures (Spectrum02695-25-s0001.pdf).** Fig. S1 and S2.

### Open Peer Review

**PEER REVIEW HISTORY (review-history.pdf).** An accounting of the reviewer comments and feedback.

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
