## [Reviewer comments · Microbiology Spectrum]

Microbiology Spectrum

Impact of synonymous mutations in the *bla*_{TEM-3} gene on gene expression and *Escherichia coli* fitness

Rinku Dhungana, Heba Kaadan, Aparna Paudel, and Peter Oelschlaeger

Corresponding Author(s): Peter Oelschlaeger, Western University of Health Sciences College of Pharmacy

Review Timeline:

Submission Date:	October 2, 2025
Editorial Decision:	October 16, 2025
Revision Received:	October 27, 2025
Accepted:	October 31, 2025

Editor: Pablo Power

Reviewer(s): The reviewers have opted to remain anonymous.

Transaction Report:

DOI: <https://doi.org/10.1128/spectrum.02695-25>

Re: Spectrum02695-25 (Impact of synonymous mutations in the *bla*_{TEM-3} gene on gene expression and Escherichia coli fitness)

Dear Dr. Peter Oelschlaeger:

Thank you for the privilege of reviewing your work. Below you will find my comments, instructions from the Spectrum editorial office, and the reviewer comments.

Based on the reviews and reviewer responses, your manuscript can be now considered as "Editorial accepted" pending final checks.

Revision Guidelines

Sincerely,
Pablo Power
Editor
Microbiology Spectrum

Re: Spectrum02695-25R1 (Impact of synonymous mutations in the *bla*_{TEM-3} gene on gene expression and *Escherichia coli* fitness)

Dear Dr. Peter Oelschlaeger:

Your manuscript has been accepted, and I am forwarding it to the ASM production staff for publication. Your paper will first be checked to make sure all elements meet the technical requirements. ASM staff will contact you if anything needs to be revised before copyediting and production can begin. Otherwise, you will be notified when your proofs are ready to be viewed.

Sincerely,
Pablo Power
Editor
Microbiology Spectrum